# The Effect of Perceptual Optimization on Color Space Navigability

Trevor D. Canham[*]
York University

I. Scott MacKenzie[†]
York University

Richard F. Murray [‡]
York University

Michael S. Brown [§]
York University

## ABSTRACT

An experiment with 22 participants examined the effect of perceptual optimization on color space navigation in a direct manipulation task. Participants made ten color matches using a 2+1 degrees of freedom trackball to control hue, saturation, and value in the HSV color space and hue, chroma, and intensity in the HCI color space. (HCI is an IPT-based color space.) Participants completed the task significantly faster with the HSV space, at 41.3 s per color pair, compared to 46.7 s with HCI. However, the difference in accuracy between color spaces was not significant. Correction path analysis suggests that the benefit of HSV's uniform geometry outweighed HCI's perceptual optimization. This was demonstrated by a significant difference in the number of times participants encountered gamut boundaries, which for HSV was about 0.5 times per trial versus 1.5 for HCI. Participant responses and comments revealed that a separate control for the hue attribute was beneficial for navigating the color space.

**Index Terms:** Color spaces, Color Correction, Color Appearance, Psychophysics, User Interfaces

## 1 INTRODUCTION

In image and graphics editing suites, color manipulation tools are among the most difficult to learn and operate. An explanation is that color perception involves complex spatiotemporal dynamics, so our experience is non-linearly related to scene physics. Visual color illusions readily demonstrate such complexity, as in Figure 1. In this example, the central colored rings in the left and right patterns are physically identical (i.e., they have the same RGB code value), but appear different due to their surroundings and size (the effect of which can be observed by viewing the figure at different distances). Though often regarded as perceptual anomalies, illusions like this reveal the complicated dynamics of the visual experience.

The visual illusion from Figure 1 also demonstrates a common cause of frustration for users of standard, ubiquitous color picker interfaces. Users pick a color and find it looks incorrect when viewed in context [17]. For this reason, it is advantageous to design tools to manipulate colors directly via a hardware interface to allow simultaneous color evaluation within context, avoiding the need for an additional visual influence of a software GUI.

Figure 1 exemplifies why such a tool is needed. In asymmetric matching experiments, psychophysicists use such patterns to explore perceptual dynamics by asking participants to adjust the central ring in the test patch (right) to match the target (left) [5, 26]. The physical difference between the participant response and the target sums up the effect of the pattern arrangement on viewer perception and can be used as training data for visual models. To do the task, participants need a tool to manipulate the test patch while limiting their visual input to the displayed patterns. A successful tool requires a control-display (CD) transfer function, which allows users to make

---

[*]e-mail: tcanham@yorku.ca

[†]e-mail: mack@yorku.ca

[‡]e-mail: rchrd.mrry@gmail.com

[§]e-mail: mbrown@eecs.yorku.ca

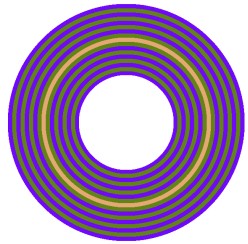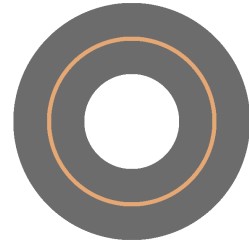

Figure 1: Shevell rings visual color illusion. The central ring on the left is physically identical to the central ring on the right but appears different due to visual adaptation to the surrounding rings.

perceptually relevant and consistent adjustments and learn a sensory-motor mapping to quickly orient themselves in the color space.

This work presents a user experiment that compares the legacy HSV (hue, saturation, value) color space with the HCI (hue, chroma, intensity) color space. HCI is a perceptually optimized (PO) color space that fits psychophysical data to eliminate contamination between perceptual attributes (e.g., hue, saturation, brightness, etc.). The goal of HCI is to improve the ease of use for a simple color-matching task by reducing confusion related to the effect of control axes on color appearance. However, the decorrelation of these attributes introduces geometric irregularity at the boundaries of the adjustment space which is an additional source of confusion. The primary contribution of this work is to evaluate this trade-off by comparing HCI to HSV, whose perceptual attributes are poorly decorrelated but has smooth boundaries. The geometric properties of the color spaces and participants' ability to navigate them are evaluated based on task completion time and accuracy. These measures are relevant to the psychophysics application as the former limits the number of trials that can be reliably conducted and the latter limits the confidence of the results.

In the following sections, background on color spaces and related work on color manipulation interfaces will be presented. Then, the user study methodology will be described, followed by the results and discussion on the comparative performance of the tested spaces and the beneficial characteristics of a color space for direct manipulation. Finally, the conclusions of the work will be summarized, and future work will be proposed.

## 2 BACKGROUND

Color manipulation requires a color coding system. Describing all colors and their ordering relative to one another expands this to a "volume" or "space" of colors with particular geometric properties. Over the years, several spaces have been proposed to account for perceptual dynamics. Readers may be familiar with RGB, HSV, or CMYK, which are commonly implemented in imaging and graphics applications. These are popular for their ease of use but are device-dependent as their values correspond to the drive values of the primary colors of a particular device and would result in a physically

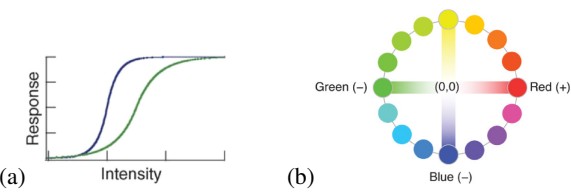

(a)     (b)

Figure 2: General components of PO color spaces. (a) Depiction of the response non-linearity of retinal neurons at two different mean light intensity levels. Figure from Cyriac et al. [8] adapted from Dunn, and Reike [10]. (b) Opponent color representation diagram, where unique hue pairs yellow/blue and red/green are placed on opposite ends of the two axes. This representation reflects the phenomenon that additive combinations of other hues can not obtain these axes' endpoints and that opponent colors cancel each other out (become gray) when mixed. Figure from Shevell [31].

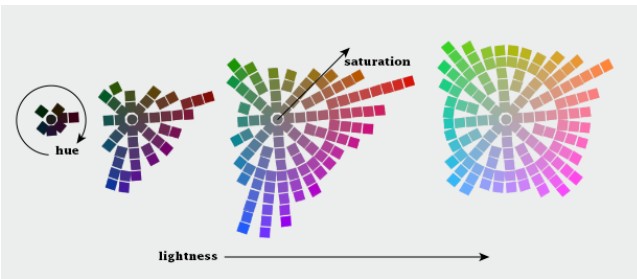

Figure 3: Graphical depiction of hue, saturation and lightness axes. Hue is represented by the angle about the midpoint, saturation is represented by the distance from the midpoint, and different lightness planes are shown side by side. Figure from NASA [32].

and perceptually different stimulus outside of this context. For this reason, they have a low perceptual correlation. On the other hand, there exist color spaces that process objective physical light measurements through mechanisms similar to those found on the visual pathway, such as neural response non-linearity and opponent color processing (see Figure 2).

Perceptually optimized color representations allow image processing tools to better correlate with the human perception of scenes and objects captured in images. In terms of functionality, these color spaces improve on representations that do not account for perceptual processes in two ways. The first is *perceptual linearity* (PL) - meaning that different perceptual attributes (hue, saturation, lightness/brightness, etc.) can be separated from one another, as demonstrated in Figure 3. The second is *perceptual uniformity* (PU) - meaning that Euclidean distance in the space corresponds to perceptual distance throughout the color volume (e.g., a just noticeable difference, JND, for Euclidean distance is the same between two colors in the red region as in the blue region).

Throughout this paper, a number of color-appearance terms will be used. While each term has a unique definition (as each color space channel has a unique mathematical definition), some terms are used interchangeably and are similar enough to be compared in this context. In particular, the saturation and value channels of HSV will be compared to the chroma and intensity channels of HCI, respectively. The analogous terms of brightness and lightness can also be equated to value and intensity. The interested reader can find a more in-depth breakdown of color terminology in Fairchild's *Color Appearance Models* [14].

In the context of direct color manipulation tools, both of the

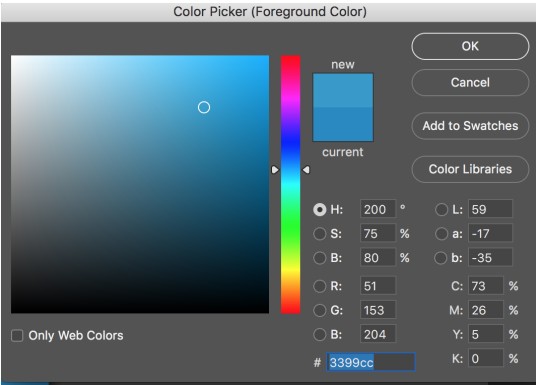

Figure 4: Screenshot of the color picker interface in Adobe Photoshop CC 2018.

main properties addressed in PO may have an impact. In a space optimized for PL, it is easier for users to make multi-dimensional adjustments by matching recognizable perceptual attributes one at a time instead of simultaneously balancing between several degrees of freedom (DOF). Conversely, with poor PL, the attributes are not cleanly separated, so matching one attribute and later correcting another would result in a shift in the first property. This requires users to engage in a frustrating adjustment loop to center on the intended target. If the navigation space has good PU, users can learn a consistent motor-sensory relationship between their interaction with the control and the displayed shift. With poor PU, users may need help learning such a relationship, as their controls behave irregularly in different color regions. In summary, both properties have the potential to affect a user's ability to quickly familiarize themselves with a direct color manipulation tool.

## 3 RELATED WORK

Recent works on color manipulation interfaces cite that color picker tools (like the one shown in Figure 4) have remained largely ubiquitous and unchanged over the last 30 years. Most of this work focuses on improving color picker interfaces with respect to their ergonomics, swiftness, and accuracy [11, 15]. In the work of Ebbinason and Kanna [11], a user study is conducted to verify the efficacy of a touchscreen color picker interface, where participants are asked to match a series of reference colors. The work of Jalal et al. [17] provides an in-depth overview of the cross-disciplinary use of color selection tools and proposes a series of new tools to address different color manipulation needs. Bauersfeld and Slater analyze existing color manipulation tools and offer a speculative vision for improving them, including using perceptually uniform color representations [1]. Ottosson proposes a PO color picker and outlines useful design criteria for a color space designed for a color picker tool [27]. These properties include a simple geometric shape to display in the interface (e.g., cylindrical representation), the most saturated color of a given hue at the edge of the interface for easy access, perceptual quantization (each code value is at or below JND with respect to its nearest neighbors), PL and PU. While several of these works take for granted the necessity of PO spaces for color manipulation tools, none test directly for the effect of PO on usability.

The most analogous works to the experiments proposed here are those of Schwarz et al. [30], and Douglas and Kirkpatrick [9]. In the former, an extensive battery of color spaces was tested for ease of navigation. Using a tablet-based interface, participants completed a simple color-matching task. The results showed that using the RGB color space, matches could be made quickly but less accurately, and vice versa using HSV. Several other spaces were tested, but the

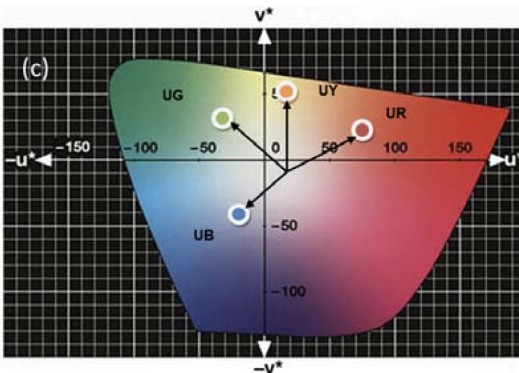

Figure 5: Color navigation with axes orientated towards unique hues. Figure from Chauhan et al. [6].

attributes represented by the color channels differed between them. For this reason, a direct comparison of the effect of PO cannot be made using these data. Douglas and Kirkpatrick expand on this work by testing the two spaces with the greatest differentiation (RGB & HSV), but do so in two configurations – one employing a software interface illustrating to participants their current location on each axis and one without. Their experiment revealed that the effect of including an interface dwarfed the effect of color space.

### 3.1 Color Interfaces for Psychophysics

In many psychophysical experiments involving color manipulation, custom interfaces are designed to limit the navigable range in color space and simplify the experimental task. In most cases, this relies on the assumption that the effect being probed can be approximated as a change in just one or two color appearance attributes, allowing for a reduction in the dimensionality of the control interface [4,6,26]. The work of Chauhan et al. [6] is centered around a user experiment testing the efficacy of a control interface simplification, given the task of navigating to an achromatic (gray) setting from a chromatic (colored) starting point. In this work, the axes of the control space are oriented towards experimentally defined "unique" hues (red, green, blue, yellow), to assist users in their directional navigation of a 2D chromaticity space. The authors found that this navigation style resulted in greater consistency of user responses compared to a navigation scheme oriented to the native axes of the CIELUV color space [7]. Unfortunately, as can be intuitively assumed from Figure 5, achieving this redefinition of cardinal axes would require warping the chosen color space, leading to losses in both PU and PL.

### 3.2 Perceptually Optimized Color Spaces

A number of psychophysical experiments have examined color spaces with respect to PL and PU, using different tasks and contexts (e.g., real-life objects versus images on the screen) [3, 12, 16, 18, 24, 25, 33, 34]. A visualization of the data in these experiments is presented in Figure 6. Several in-depth surveys on PO color representations have recently been published [22, 29, 34]. In the work of Lissner and Urban, the engineering trade-offs in PO are clearly outlined [22]. The authors note that PL and PU are at odds due to Judd's paradox of hue differences [19]. If the hues are in an angular arrangement about a midpoint and extend linearly outward, as in Figure 3, the color difference from one hue value to the next in the central region is not equal compared to the edges. They also point out interesting trade-offs within each of the criteria of PO. First, the decorrelation of two perceptual attributes often comes at the cost of the correlation of the others. Also, optimizing PU for threshold

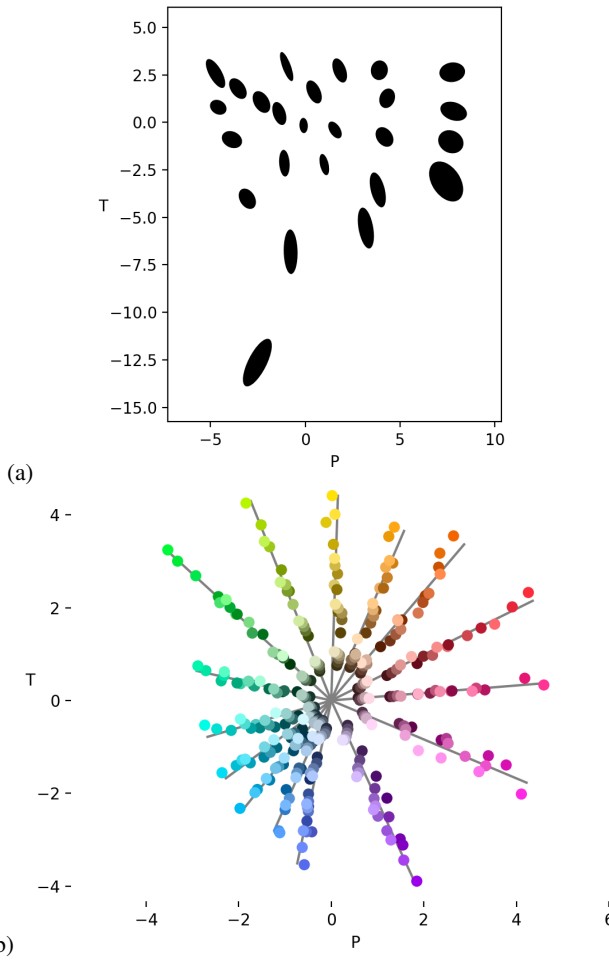

(a)

(b)

Figure 6: Psychophysical data for perceptual optimization of color spaces, plotted in the chromaticity channels P and T of IPT. (a) MacAdam color difference ellipses [24]. The ellipses represent one experimentally derived JND about the central point. The PU of a given color space can be evaluated by the consistency of size and shape of these ellipses. (b) Hue linearity from Ebner and Fairchild [16]. To derive these data, observers match the hue of color patches with different saturation levels, allowing for linear plots to be constructed which maintain hue as saturation is modulated.

(small) color differences can misrepresent supra-threshold (large) color differences. The authors conclude that these trade-offs reflect the fundamental limitations of representing color with Euclidean geometry.

In general, color spaces are compared in these works based on their usefulness for image processing tasks or fitting psychophysical data. There is little research on the qualitative experience of how well they meet optimization goals, and to our knowledge, no study has examined ease of navigation. Potential explanations are that navigation involves the perceptually relevant temporal dimension, which none of the spaces addressed here account for, and the large degree of noise and variability resulting from 3DOF method-of-adjustments tasks.

Below, a collection of PO color representations will be briefly compared. While this does not represent an exhaustive list of the existing representations, those highlighted here are the most commonly tested in topic surveys due to their long history of use or exceptional performance in predicting psychophysical data. All

Table 1: Pugh matrix comparison of prominent color spaces in terms of linearity, uniformity, popularity, complexity, and range. '+' represents positive features (e.g., lower complexity receives a '+' marking)

| Color Space | PL | PU | Popularity | Complexity | Range |
|---|---|---|---|---|---|
| CIELAB [7] | + | - | + | + | - |
| IPT [12] | + | - | + | + | + |
| $LAB2000_{HL}$ [22] | + | + | - | - | - |
| $J_za_zb_z$ [29] | + | + | - | + | + |
| CAM16-UCS [23] | - | + | - | - | + |
| HSV | - | - | + | + | - |

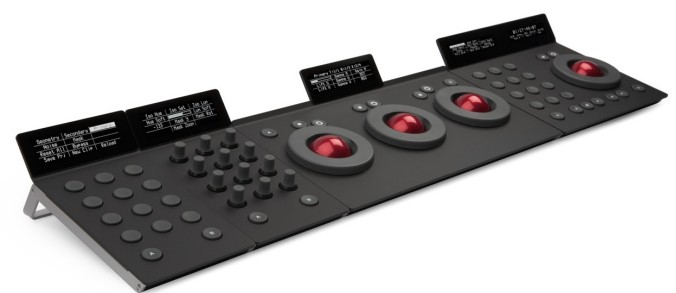

Figure 7: Tangent *Element* color correction interface (tangent-wave.co.uk). Trackballs are heavily used in cinema and television color correction.

Table 2: Absolute CIEXYZ values of ten color pairs, randomly sampled throughout an example color gamut (where Y is display luminance in $cd/m^2$).

| Starting Point | | | Target | | |
|---|---|---|---|---|---|
| X | Y | Z | X | Y | Z |
| 92.2 | 145 | 36.6 | 48.5 | 42.4 | 112 |
| 27.2 | 19.1 | 39.0 | 71.3 | 74.0 | 9.09 |
| 101 | 144 | 43.8 | 59.6 | 32.4 | 160 |
| 16.5 | 10.0 | 7.71 | 124 | 122 | 148 |
| 101 | 120 | 27.8 | 82.6 | 54.9 | 133 |
| 137 | 144 | 118 | 65.3 | 32.0 | 143 |
| 72.0 | 83.1 | 170 | 75.5 | 108 | 75.9 |
| 108 | 141 | 73.2 | 104 | 83.5 | 113 |
| 83.6 | 130 | 82.8 | 82.9 | 104 | 109 |
| 127 | 120 | 176 | 101 | 86.6 | 97.9 |

spaces presented consist of three dimensions – one corresponding to lightness/brightness and two corresponding to opponent color axes. The perceptual attributes of hue and saturation can be easily extracted from the opponent axes via conversion to polar coordinates. In addition to the phenomenon of opponent color channels, all of these spaces start with measurements of physical energy and include a non-linear function, approximating the response of retinal neurons.

- Recommended in 1976 by the Commission Internationale de l'Éclairage (CIE), CIELAB is regarded as the most widely used PO space [7]. While the space has good separation between achromatic and chromatic axes, it is not free of cross-contamination between saturation and hue, with strong deviations in the red and blue regions [20]. Visual experiments also illustrate its lack of perceptual uniformity [2]. Based on CIELAB, the most widely used metric for color differences is $\Delta E_{00}$, due to its high PU. This metric will be employed for the remainder of this work.

- IPT is a computationally simple space presented by Ebner and Fairchild to fit the authors' data on hue linearity [12]. As a result, it has a strong decorrelation between hue and saturation but suffers from cross-contamination between achromatic and chromatic axes [21]. "I" represents the achromatic intensity channel which correlates to brightness, while "P" and "T" are chromatic opponent channel axes, named after the red-green and yellow-blue retinal responses "protan" and "tritan."

- Lissner and Urban present a re-optimization scheme to fit existing color spaces to PU and PL data and derive an updated version of CIELAB ($LAB2000_{HL}$) where its performance is improved on both criteria [22].

- Another recent space called $J_za_zb_z$ was also optimized for a balance between PL and PU but used psychophysical data which cover a greater range of light levels, to future-proof its validation for new display formats which present images with greater contrast and saturation range [29].

- CAM16-UCS [23], offers strong PU, but suffers from cross-contamination between achromatic and chromatic axes and between hue and saturation.

In addition to PO, computational complexity, popularity of use, and validation range are factors in selecting the most relevant color representations to test. A simple binary comparison based on these criteria is presented in Table 1, conglomerating the experimental results of the publications cited above.

While a comparative study of the navigability of all of these spaces would be helpful in designing direct color manipulation tools for psychophysical experiments and image editing suites, it is beyond the scope of this work. Instead, the question can be narrowed to component parts. First, does PO significantly affect the navigability (speed and accuracy of use) of direct color manipulation tools? Second, since PL and PU are a trade-off, which is more relevant for the design of these tools?

The following sections describe an experiment that targets the concept of PL by comparing HSV (poor PL) with a color representation referred to as HCI (hue, chroma, intensity) derived from the IPT space (good PL). The HCI space is derived simply by converting P and T channels into a polar representation, resulting in Hue (angle) and chroma (magnitude) correlates, and the I channel is used directly. Following the example of Schwarz et al. [30], a 2+1 DOF control interface was employed, where saturation/chroma and value/intensity are controlled with 2 DOF, and hue is given a separate 1 DOF control. Figure 7 shows a standard input device for color correction, consisting of multiple 2+1 DOF trackballs. Besides sensing left/right and forward/back rotation, the device also senses lateral rotation of the ball [13]. Given that it is conducive for fluid and precise color manipulation, participants will employ this hardware interface to complete a symmetric color matching task between a set of ten color pairs randomly sampled throughout the RGB gamut. The colors are introduced in Table 2.

Figure 8 shows an example RGB color gamut plotted in HSV and HCI spaces. Since the HSV space has no absolute physical definitions, its boundaries are relative to the display primaries. In contrast, IPT is not limited to the display color gamut, so the monitor gamut represents a subset of the space and takes on a particular shape. The resulting volume of the HCI space features jagged, irregular edges. This jagged shape is potentially problematic for a color navigation space as controls react irregularly when users encounter gamut boundaries (i.e., the color on the screen may stop changing as users go out of bounds.) Other color representations like $J_za_zb_z$ were explored during preliminary testing (by similarly converting opponent channels $a_z$ and $b_z$ to polar coordinates to derive hue and chroma channels $H_z$ and $C_z$), but were found to be virtually impossible to navigate due to geometric properties like the compression of the

space around red hues shown in Figure 8c. CIELAB, $LAB2000_{HL}$, and CAM16-UCS also had similar drawbacks when converted to represent the appearance attributes.

Assuming a common strategy of first matching the hue value of the test patch to match the target and later correcting saturation/chroma and value/intensity parameters to perfect the match, one can calculate an ideal path distance in $\Delta E_{00}$ units in both spaces as a pre-experiment analysis. In Figure 9, "ideal" paths for an example color set are compared between the two spaces. Averaged over the ten color pairs, the paths tend to be more direct in HCI space, at an average of 213 $\Delta E_{00}$, compared to the paths in HSV, which sum to 283 $\Delta E_{00}$ on average.

Another benefit of using the HCI space can be observed in Figure 10. Here, the starting and target patches are compared to the results from HCI and HSV, where the hue channel is perfectly matched to the target, but saturation/chroma and value/intensity channels remain at the starting point. In these examples, it can be observed that the HCI hue adjustment is better decorrelated with the remaining two channels, as the perceived intensity and chroma stay closely matched to the starting point. HSV, on the other hand, exhibits large shifts in perceived saturation and brightness with shifts in hue.

## 4 METHOD

### 4.1 Participants

The experiment included 22 participants (15 M, 7 F) aged 21-50. The participants included students and staff from York University. All participants had normal color vision verified by an Ishihara color test, and were unaware of the purpose of the experiment. While a small handful of participants had experience interacting with color spaces in the context of image processing, only one participant had prior experience performing color matching tasks. They received a small gift as an incentive for their participation.

### 4.2 Apparatus

The experiment used a late 2018 MacBook Pro with software coded in MATLAB using the Psychtoolbox [28]. The software managed the cadence of the experiment, displaying stimuli, querying the input device, and saving participant performance data. A Dell P2317H monitor was employed as a display, with specifications listed in Table 3.

The display output was measured for primary colors and white at 16 drive values with a PhotoResearch PR-655 spectrophotometer to verify additivity and consistency in the color space. The display's gamma function and primary values were derived from the measures to accurately represent the colorimetry (outlined in Table 2) and ensure that the recorded color values along adjustment paths accurately reflect the physical stimuli participants were viewing.

The experiment was conducted in a dark environment for the greatest degree of control and repeatability. The test and target stimuli each subtended a visual angle of $2°$. To impose consistency for the participant's contrast adaptation, a Gaussian noise layer (visible in Figure 10) was added to each color channel of the background, test, and target, with the mean set to the reported colorimetry values in Table 2. The mean chromaticity of the background was set to the monitor white point at a luminance of 32 cd/m$^2$.

A Kensington SlimBlade trackball was used as an input device (Figure 11). The trackball features 2+1 DOF control via forward/backward rotation, left/right rotation, and, additionally, lateral rotation. The CD gain level was normalized for each color space by its gamut volume (the 3D solid of colors reproducible by the display) to account for their different numerical scales.

### 4.3 Procedure

At the start of each experiment session, participants were welcomed to the lab and seated at the experiment workstation. First, an Ishihara color test was administered to screen out color-blind participants.

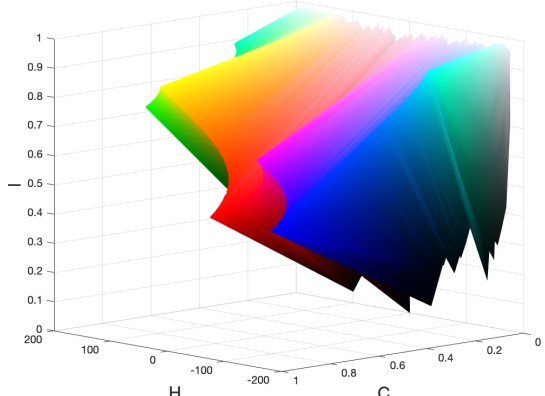

(a)

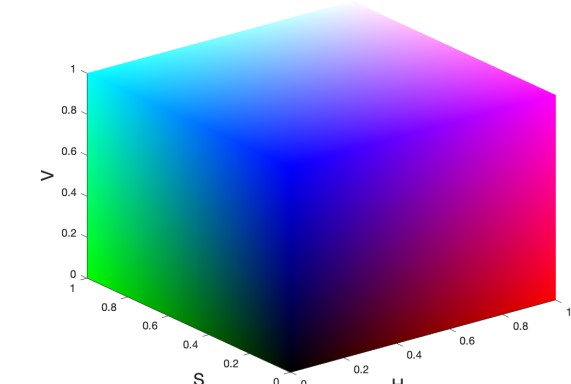

(b)

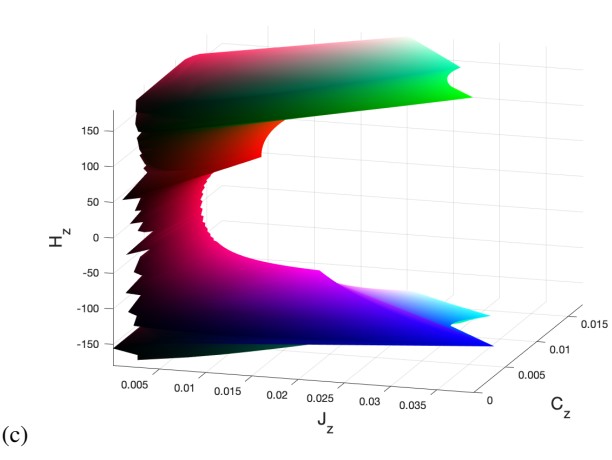

(c)

Figure 8: Dell P2317H LCD monitor gamut plotted in (a) HCI color space (b) HSV color space and (c) $H_zC_zJ_z$ color space. Since HCI is an absolute measurement and is not relative to a display gamut, it takes on a jagged shape in comarison to HSV. Other PO spaces like $J_za_zb_z$ have geometric qualities that reduce navigability when converted to perceptual attribute representations, like the stark reduction in volume observed in the mid-range hue values of (c).

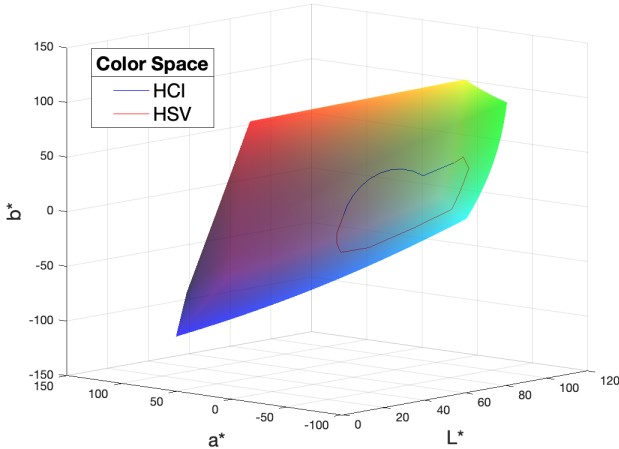

Figure 9: Ideal correction paths in the context of the RGB gamut for color pair #8, plotted in CIELAB space, for both HCI and HSV spaces. Note that using the HCI space, users should be able to approach the result more directly in this example.

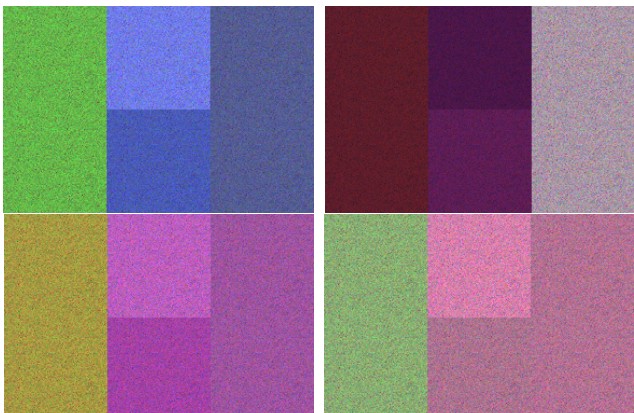

Figure 10: Four example hue matches, where starting and target colors are shown on the right and left sides, and hue matches for each space are shown in the middle – HCI on top, HSV on the bottom. Note that when matching the hue of the starting color to the target, HCI better maintains the saturation and brightness compared to HSV.

Table 3: Dell P2317H monitor specifications.

| Dimensions | 50 x 30 cm |
|---|---|
| Viewing Distance | 72 cm |
| Peak Luminance | 179 cd/m$^2$ |
| Gamma | 2.24 |
| Primaries | x,y |
| Red | 0.652, 0.338 |
| Green | 0.325, 0.618 |
| Blue | 0.156, 0.044 |
| White | 0.312, 0.333 |

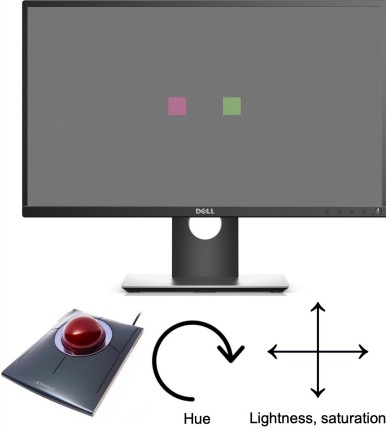

Figure 11: Experimental apparatus, including the Dell display and trackball interface.

Then, the experimental instructions were read aloud, detailing the operation of the experimental interface, the cadence of the experiment, and the experimental task. Participants were informed that they would see two patches on the screen, the "target" patch on the left and the "test" patch on the right. The goal was to adjust the color of the test patch to match the target patch and to proceed at a quick but comfortable pace. Participants were encouraged to move on if they spent too much time making a fine adjustment. Printed diagrams of Figures 3 and 11 were shown to demonstrate the control interface.

Then, participants were asked to complete a training session to introduce the experimental interface. A series of color names were listed on the screen, and participants were asked to navigate to each of the colors. Participants started the body of the experiment after completing the training session. Throughout the experiment, participants performed ten color pair matching tasks for two color spaces. The color pairs were presented in random order. After completing ten matches for one space, they performed the ten matching tasks with the other color space. An equal number of participants started on each color space to account for learning effects. Participants completed the experiment in about 20 minutes.

### 4.4 Design

The user study employed a $2 \times 10$ within-subjects design with the following independent variables and levels:

- Color space (HSV, HCI)

- Color pair (1, 2, ... 10)

Color pairs were distributed throughout sRGB gamut.

Other factors potentially influencing results were held constant, including the background (mid-gray) and environment (dark).

The dependent variables were task completion time (averaged over the ten color pairs for each color space), accuracy ($\Delta E_{00}$ color difference between target patch and final participant response), and gamut encounters (the number of times participants encountered the monitor gamut boundaries.) The task completion time and response accuracy averaged, over a diverse participant pool, and uniformly distributed color pair sets reflect the difficulty of the interface. The total number of trials was 440 (= 22 participants $\times$ 2 color spaces $\times$ 10 color pairs).

(a)

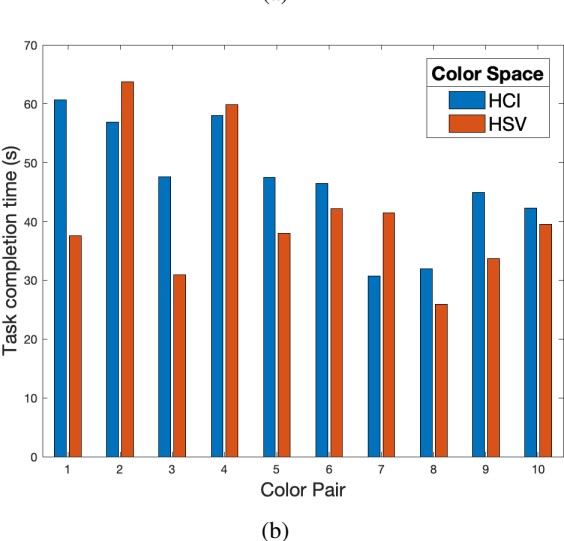

(b)

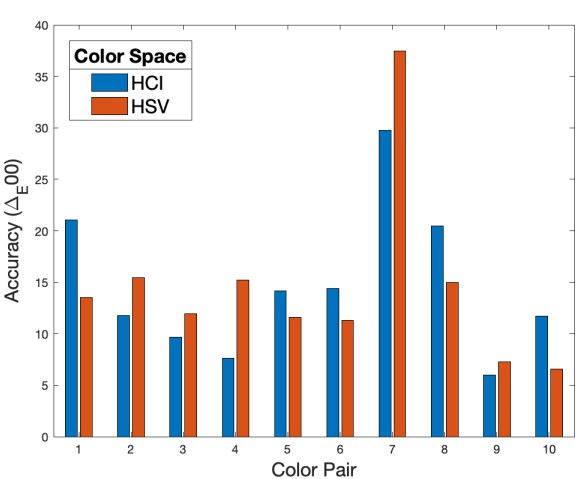

Figure 12: Results for color pair and color space. (a) Task completion time, (b) Accuracy.

## 5 RESULTS

The overall mean for task completion time was 44.0 seconds (s). HSV was faster at 41.3 s than HCI at 46.7 s. The main effect of color space on task completion time was statistically significant according to an N-way ANOVA test ($F_{1,21} = 4.65$, $p < .05$). The overall standard deviation was 33.9 s. The grand mean for accuracy was 14.6 $\Delta E_{00}$, but the effect of color space on accuracy was not significant ($F_{1,21} = 0.01$, ns). The overall standard deviation was 21.5 $\Delta E_{00}$.

Analyzing correction paths, participants encountered the monitor gamut boundaries roughly once per trial. Compared to HSV, where participants only encountered gamut boundaries 0.5 times/trial, the HCI space encounters were greater at 1.5 times/trial. The Wilcoxon Signed-Rank test shows a significant effect of color space on gamut encounters ($z = -8.084$, $p < .001$).

As shown in Figure 12, the independent variable of the color pair had a significant effect on task completion time ($F_{9,21} = 6.10$, $p < .001$), as well as accuracy ($F_{9,21} = 7.15$, $p < .001$).

In addition to the observations above, the ANOVA showed partic-

(a)

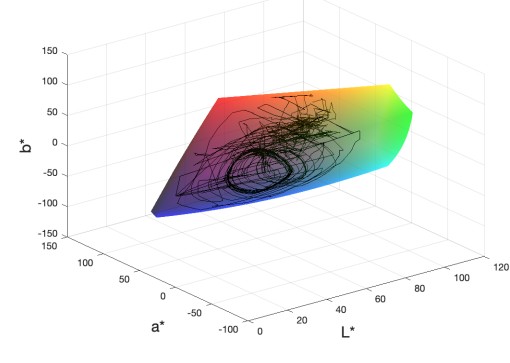

(b)

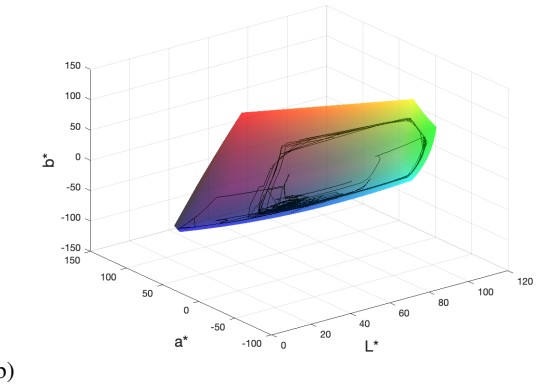

Figure 13: Example participant correction paths for color pair #2 in (a) HCI and (b) HSV spaces.

ipants differed significantly in task completion time ($F_{21,19} = 13.37$, $p < .001$) and accuracy ($F_{21,19} = 3.10$, $p < .001$). Figure 13 shows a series of example correction paths, where all participant paths for a particular trial are overlaid within the monitor gamut. It can be observed that participants used a wide variety of strategies to reach the intended target, and in some cases, they collectively traversed nearly every region of the monitor gamut.

Figure 14 demonstrates the speed-accuracy trade-off for an example color pair. Despite significant variability, the least accurate trials were among the fastest completed and trials with the longest completion times were among the most accurate.

In Figure 15, the average $\Delta E_{00}$ color distance from the target color is plotted from trial start to finish. It can be seen that participants approached the target color in a roughly linear fashion. Also, the max-normalized cumulative $\Delta E_{00}$ correction distance per color channel, averaged over all participants and trials, is plotted in Figure 16. This demonstrates that most of the color distance traversed by participants was covered using the hue control, particularly at the beginning of each trial, and gradually less as the trial continued.

## 6 DISCUSSION

While the task involved a speed-accuracy trade-off (i.e., more accurate matches generally required more time for adjustment), the significant statistical effect of color space on time and its non-significant effect on accuracy indicates that participants were able to complete the task more easily in HSV. In spite of the shorter "ideal" path lengths presented in Figure 9 and the perceptual linearity of HCI demonstrated in Figure 10, the geometric irregularity of the space, shown in Figure 8, resulted in a greater number of encounters with monitor gamut boundaries. These encounters could have affected

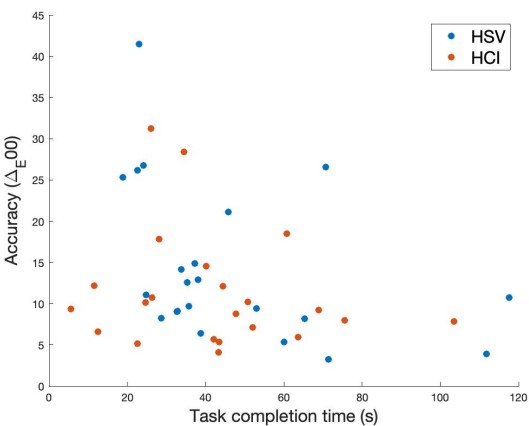

Figure 14: Task completion time versus accuracy for both color spaces, color pair #6.

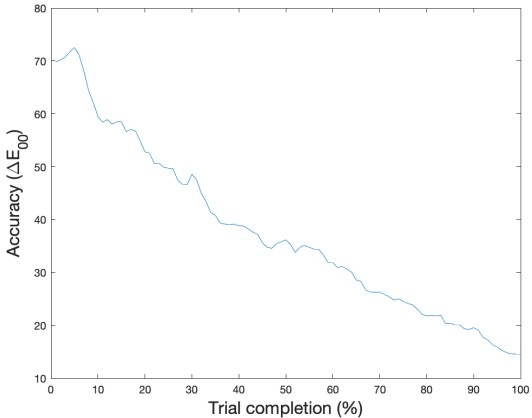

Figure 15: Analysis of the $\Delta E_{00}$ distance travelled over the course of a trial, averaged over both color spaces and all observers.

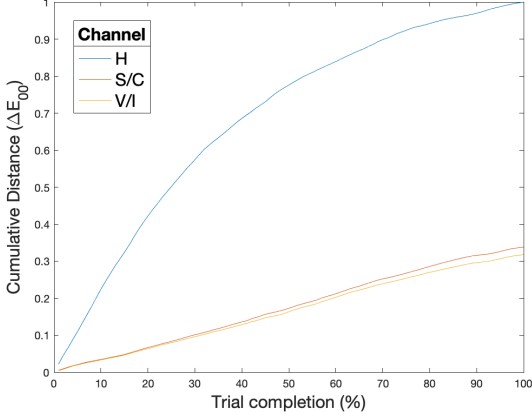

Figure 16: Cumulative channel wise $\Delta E_{00}$ distance travelled over the course of a trial, averaged over both color spaces and all observers and normalized by the maximum distance.

the participants' ability to quickly complete the task by requiring them to proceed with caution to stay in gamut and make an extra effort to navigate the boundaries.

The strong effect of color on the dependent variables is not explained by the initial $\Delta E_{00}$ distance between target and test patches (resulting in a correlation $R^2 = .14$ for accuracy, .02 for time). However, the mean task completion times for each color set correlate better with the sum Euclidean distance of S/C and V/I channels ($R^2 = .61$), compared to the hue distance ($R^2 = .02$). Considering accuracy, the mean Euclidean distance on hue channels correlates slightly better than the initial $\Delta E_{00}$ distances ($R^2 = .18$), but this correlation is still poor. This indicates that the difference in task performance between color pairs could be specifically attributed to the difference in saturation/chroma and intensity/value between starting and target colors and that it was generally easier for participants to make quick hue matches, regardless of the starting distance.

The experiments of Douglas and Kirkpatrick [9] and Schwarz et al. [30] made similar comparisons to the present study, employing an analogous interface scheme and task. However, completion time and accuracy differed quite a bit. With respect to time, the previous experiments had overall means of 60.9 s and 80.7 s, respectively, compared to our 44.0 s. Using the legacy color difference metric CDU (also known as $\Delta E_{76}$), the prior studies' participants matched colors within ranges of 8.23 and 7.06 CDUs, respectively, while the grand mean of our results equates to 13.2 CDUs. Schwarz et al. show a similar plot to Figure 15, but theirs showed a non-linear progression where observers spent significantly more time refining results after quickly converging on the color region. Figure 15, on the other hand, is nearly linear. This indicates that our participants did not spend as much time refining their results, which may be caused either by the coarseness of the controls or the experimental instructions, which specifically advised participants not to spend a significant amount of time making minor refinements. In contrast, the experimental instructions of the previous studies both asked observers to continue honing in on a match until it became "extremely difficult" to improve.

During informal post-experiment discussions, a common trend was observed—that participants primarily employed strategies centered around the hue control. Participants reported that this was the easiest perceptual attribute to target and match. For this reason, a number of them used a strategy of first increasing saturation such that hue angle adjustments would be more coarse and then honing in on the correct saturation/chroma and value/intensity settings after matching the hue. This general strategy is reflected in Figure 16. Participants also commented on the difficulty of making precise matches to desaturated targets caused by the ineffectiveness of hue adjustment in this region. A possible explanation for participants' heavy use of the hue adjustment is that this attribute is the primary delineation in color naming and categorization boundaries. When explaining the hue attribute to several participants, they referred to it as the "color" attribute, demonstrating the equivalence described above.

## 7 CONCLUSION

An experiment was conducted to examine the effect of perceptual optimization on color space navigation. Participants made a series of color matches using a 2+1 DOF trackball interface which controlled the saturation, value and hue of the HSV color space in one series of trials and the chroma, intensity and hue of the HCI color space in another series of trials. Hue and chroma were derived by converting the P and T channels of the perceptually linear IPT space to polar coordinates. The results showed that participants could complete the task significantly faster with the HSV color space, at 41.3 s per trial, compared to 46.7 s per trial for the HCI color space. The results did not show a significant difference in accuracy between color spaces, indicating that the HSV space was easier to navigate

overall in comparison to the HCI space.

The participant path analysis suggests that the benefit of geometric regularity in the HSV color space outweighs the benefits of the HCI space's decorrelation of the hue attribute. This was revealed by a significant difference between the number of times participants encountered gamut boundaries, which for HSV averaged roughly 0.5 times per trial versus 1.5 for HCI. The participant adjustments and comments revealed that the isolated hue attribute was beneficial for navigating the color space. In addition, the difference in task completion time correlated positively with the Euclidean distance between starting and target S/C and V/I channels, but not with the hue distance. This implies that these perceptual attributes were more difficult for participants to match in comparison to hue.

Moving forward, a useful advance would be to develop an improved HCI space, which features both perceptual decorrelation of color appearance attributes (PL) and a regular shape, regardless of the display gamut. In addition, a longitudinal study on the effect of learning in this context would be of interest. One of the goals of this study was to examine participants' ability to familiarize themselves with the geometric properties of different color spaces. While many reported developing a better understanding of the controls throughout the experiment, the lack of repeated trials and short experimental duration prohibited the analysis of learning effects. A larger-scale study could explore the effect of perceptual optimization on building familiarity with color spaces.

## ACKNOWLEDGMENTS

The authors would like to thank Philipp Urban for providing the $LAB2000_{HL}$ code base, as well as each of the participants of our user study. This work was funded in part by an NSERC discovery grant.

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
