# OpenReview forum: "The Effect of Perceptual Optimization on Color Space Navigability"
_graphicsinterface.org/Graphics_Interface/2023/Conference — GI 2023_

### Official Review · Reviewer_zQEg · 2023-01-08
**Experiment quantiative study - Color selection UI design**

**Rating:** 4
**Confidence:** 4

**Review:**

This paper examines two different color schemes while task completion time and accuracy were measured. A new user interface was introduced to improve color picking.
The paper provided a bit of context on the importance of using different color schemes; however, it did not clearly point to the gap in the literature that is filled by this work. It is unclear how the experiment design and the presented research will answer the research question/gap.
Section 3 discusses a mix of related work, research questions, and experiment design. I suggest clearly distinguishing these sections to help with reading and better understanding the research. The experiment design is not justified with the rationale and support of the literature.
The experiment results are not clearly explained. From my understanding, the accuracy did not differ between the two color scheme models, what does this mean? Will this help the future design of the color palette UIs? It was concluded that participants were able to complete the task more easily in HSV despite a non-significant result of accuracy. Minor issue: it mentioned the use of a tangent keyboard; however, the video is not using this hardware (similar but not the exact model).
Later in the discussion, the results of the experiment were compared with other studies, such as Douglas and Kirkpatrick and Schwarz et al. work; however, it is unclear if the exact same experiments were tested to compare them.

Overall, it is difficult to fully understand the experiment, study's goal, and interpret the data collected from the presented study. I suggest the following to improve the paper: clarifying the experiment set up and backing up the study setup with research, representing the results of the study, and stating how the results can be useful for future studies. In general, paper’s writing can be improved. For now, it reads similar to a project report, and effort should be made to present the work with more scientific writing standards.

---

### Official Review · Reviewer_bJ1i · 2023-01-17
**a user experiment comparing color matching performance in two different color spaces that shows that HSV is performing better than a perceptually optimised color space**

**Rating:** 6
**Confidence:** 3

**Review:**

The paper conducts an experiment that compares the performance of two color spaces for the task of color matching. One color space is an often used and well known space (HSV) with a perceptually optimised color space (HCI/IPT). The overall finding is that HSV is more "user-friendly", in the sense that the navigation (color matching) task was faster and not less accurate.

The paper is well written, easy to follow and reproducible.  The result figure (Fig 12) shows that overall average performance of the 22 participants. Here I would appreciate more details (e.g. a scatterplot, where each dot is a participant). It would be nice to show a 2D scatterplot, where x is the completion time and y is the accuracy. This could possible give a bit more insight and intuition of the experiment.

---

### Official Review · Reviewer_dcRz · 2023-01-17
**Interesting study on how different colour model structures impact navigation in colour editing tools**

**Rating:** 8
**Confidence:** 5

**Review:**

This paper reports an empirical study of how the geometric structure of 3-space colour models in common digital use has an impact of how well the user can use them to do a common colour editing task of colour matching.   in particular they compare two known models, the well-used HSV model common to most digital colour tools and a more recent HCI model developed for perceptual optimisation (PO) that was designed to overall improve ease of use in this notoriously difficult class of user tasks. The authors' findings in fact rebut the assumption that the PO model was "easier" to use: participants were faster  with the HSV  model,  suggesting that the benefit of geometric
regularity in the HSV color space outweighs the benefits of the perceptual  optimisation because navigation is more straightforward and easier to comprehend.

This ia a well designed study that provides a small but important result to  interactive colour science problems, where the model needs ti support how users understand and can manipulate common colour editing tasks in digital tools.   The background is well covered and the experimental design and reporting was robust. One can query why the authors chose these two models to investigate, but the HSV model is indeed a legacy model in most digital tools, and while it suffers some of the gamut challenges of colour spaces  these tools, it is often the model of choice. The PO mode (HCI) has an interesting characteristic in that it "decouple" shifts in hue from saturation and brightness.  So while this might not be the most common space to investigate, it manifests a principle of perceptual optimization that warrants investigation.

The intersting take of this study is that it surfaces the role of easily understandable navigation in 3space as a critical issue in how these models can support colour matching. To my knowledge this has not been looked at before.   As colour models have migrated from the somewhat static science of device colour reproduction to interactive digital tools, these issue of interacting WITH the model becomre more and more important.

My main concern with this paper relates to the novelty effect and its potential to have influence negatively the users' performance with the more unfamiliar HCI model, as the HSV is so well known. The authors do not tell us what the digital colour editing experience of the participants was. This is a potential confound and should be explicitly addressed.

---

### Meta-Review · Area_Chair_PRKR · 2023-01-19

**Recommendation:** strong accept
**Confidence:** 5

**Metareview:**

All reviewers agree that this is an interesting area tomstudy, with R1 and R2 satisfied with the study deaign and analysis. Both recommend acceptance.
R2 adds a suggestion for a visualization to show the relationship between response time and accuracy, as they point out this might need to be more effectively conveyed.
R3, however, presents a diverging opinion about the paper, stating that the experiment reporting is hard to follow and the results unclear.  Moreover, they do not understand what the actual contribution of the research is.  This is in stark contrast to R1, who acknowledges the contribution of the work is small but nonetheless important, as it pertains directly to the design of user interface tools to manipulate digital colour.
Following these comments from R 2 and R3, i recommend the authors clarify their contribution more clearly and that they describe the problem in user terms - i.e.,  why this is important to supporting interaction techniques for colour editing in digital tools. A discussion on the importance of their dependent variables - speed and accuracy - in this problem space would set their results in clearer context.